# Cranberry Polyphenols and Prevention against Urinary Tract Infections: Relevant Considerations

**DOI:** 10.3390/molecules25153523

**Published:** 2020-08-01

**Authors:** Dolores González de Llano, M. Victoria Moreno-Arribas, Begoña Bartolomé

**Affiliations:** Institute of Food Science Research (CIAL), CSIC-UAM, Nicolás Cabrera, 9, Campus de Cantoblanco, 28049 Madrid, Spain; victoria.moreno@csic.es (M.V.M.-A.); b.bartolome@csic.es (B.B.)

**Keywords:** cranberry, urinary tract infections, UTIs, uropathogenic *Escherichia coli*, UPEC, flavan-3-ols, A-type proanthocyanidins, phenolic metabolites, antiadhesive activity, probiotics

## Abstract

Cranberry (*Vaccinium macrocarpon*) is a distinctive source of polyphenols as flavonoids and phenolic acids that has been described to display beneficial effects against urinary tract infections (UTIs), the second most common type of infections worldwide. UTIs can lead to significant morbidity, especially in healthy females due to high rates of recurrence and antibiotic resistance. Strategies and therapeutic alternatives to antibiotics for prophylaxis and treatment against UTIs are continuously being sought after. Different to cranberry, which have been widely recommended in traditional medicine for UTIs prophylaxis, probiotics have emerged as a new alternative to the use of antibiotics against these infections and are the subject of new research in this area. Besides uropathogenic *Escherichia coli* (UPEC), the most common bacteria causing uncomplicated UTIs, other etiological agents, such as *Klebsiella*
*pneumoniae* or Gram-positive bacteria of *Enterococcus* and *Staphylococcus* genera, seem to be more widespread than previously appreciated. Considerable current effort is also devoted to the still-unraveled mechanisms that are behind the UTI-protective effects of cranberry, probiotics and their new combined formulations. All these current topics in the understanding of the protective effects of cranberry against UTIs are reviewed in this paper. Further progresses expected in the coming years in these fields are also discussed.

## 1. Introduction

Urinary tract infections (UTIs) are the second most common type of bacterial infections worldwide, behind otitis media. UTIs incidence peaks in individuals are in their early 20s and after age 85 [1]. The lifetime risk for acquiring a symptomatic UTIs is about 12% in men and 50% in women, with a rate of recurrence after six months of about 40% [2]. Approximately 20–30% of young women will have a recurrent UTIs, named relapses, which can be caused by the same microorganism or by a different microorganism. Microbial host-associated reservoirs at the underlying bladder tissue or gastrointestinal tract could cause reinfection, even after an intensive treatments and subsequent negative urine culture [3]. All of this means that UTIs account for several millions of outpatient hospital visits and millions of emergency room visits with a large cost for the Primary Care and Public Health systems affecting the national economy [4].

According to the anatomical location of the bacteria, UTIs are categorized as pyelonephritis and kidney infection when they affect the upper part of the urinary tract (ureters and kidney parenchyma), and as cystitis and urethritis when they affect the bladder or urethra (lower infection tract). Clinically, uncomplicated UTIs affect healthy individuals without urinary tract anomalies, while UTIs associated with factors that compromise the urinary tract or host defense are referred as complicated UTIs, including urinary obstruction or retention caused by neurological disease, renal failure or transplantation, pregnancy and the presence of foreign bodies such as kidney stones, indwelling catheters or other drainage devices [5]. Among the most common symptoms of UTIs as cystitis, there is a frequent and urgent need to urinate and pain or burning sensation in the urethra during urination. Urine is usually cloudy, sometimes even pink color due to the presence of blood. If infection rises to the kidneys, patients could also be suffer fatigue, fever, nausea and muscle and abdominal pain [6]. Pyelonephritis is usually a more serious problem as it can lead to irreversible kidney damage or sepsis [7].

Uropathogenic *Escherichia coli* (UPEC) is the primary etiological agent for the majority of these infections, causing around 85% of cystitis, but other Gram-negative bacteria, such as *Klebsiella pneumoniae*, and some Gram-positive cocci, such as some staphylococcal and enterococcal species, seem to be also implicated in the etiopathogeny of the remaining infections [8,9]. Polymicrobial infections involving Gram-positive bacteria may be more prevalent than previously appreciated and may also impact on UTI output. As routine care, UTIs were diagnosed and treated based on symptomatology without performing a urine culture but that UTI empiric treatment commonly led to a UTI misdiagnosis [10].

Conventionally, the use of antibiotics to treat this pathology has claimed to be very operative, but they can cause prevalence of resistance among uropathogens and other adverse side effects, such as damage in intestinal microbiota. For these reasons, there is a growing interest in the search of natural therapies for UTI prevention and treatment to face increasing bacterial resistance to antibiotics and high recurrence rates [9,11]. To prevent such infections, different alternatives, such as the use of antiadhesive components, probiotics or vaccines, have been investigated.

Although the consumption of cranberries (*Vaccinium macrocarpon*) has been extensively recommended for UTIs prophylaxis and relief of adverse symptoms, the UTIs preventive activity of cranberry has been debated in the literature and numerous clinical studies have been carried out, including some recent meta-analyses [4,12,13]. Several studies have shown a protective effect of cranberry against UTIs [14,15,16,17,18]; nevertheless, others have not found significant effects [19,20]. This current controversy about conflicting results of the clinical and cost effectiveness of cranberry supplements has been attributed to different manufactured cranberry based products and doses, as well as a lack of systematic protocol for the selection of subjects and clinical assay [13].

The red cranberry is rich in several groups of flavonoids, particularly proanthocyanidins, anthocyanidins, and flavonols, together with phenolic acids and benzoates [21,22]. Among other possible mechanisms behind the protective effects of cranberries against UTIs is the capacity of cranberry polyphenols to act as antiadhesive agents in preventing/inhibiting the adherence of pathogens to uroepithelial cell receptors, which appears to be a major step in the pathogenesis of these infections [23]. Cell culture methodologies have also evidenced anti-adhesive capacity against the UPEC of flavonoids present in cranberry and their metabolites and of urine samples collected after cranberry consumption extracts [24,25,26,27,28].

Probiotic bacteria are considered another promising therapy in UTI prevention and treatment [29,30], and many human intervention studies have evaluated if the consumption of specific strains, such as *Lactobacillus* spp. can prevent or treat UTIs [11,31]. Moreover, the combination of cranberry with some probiotic strains has also been proposed to be effective for the management of recurrent urinary tract infections [32].

In the light of recent scientific publications on the matter, this review tends to recapitulate all these aspects related to protective effects of cranberry against urinary tract infections. After a brief presentation of UTI pathogenesis (Section 2) and UPEC and other uropathogenic bacteria (Section 3), the review covers the main aspects related to the use of cranberries in UTI prophylaxis (Section 4). Particular attention is given to the antiadhesive activity derived from cranberry consumption that has been long considered as a main mechanism involved in the protective effects of cranberry against UTIs (Section 5), and to other potential mechanisms involved in cranberry efficiency against UTIs, including interaction with gut microbiota and renal metabolism (Section 6). Finally, the potential of probiotics against UTIs and their combined action with cranberries are also considered as a promising strategy for future treatments against UTIs (Section 7). Global conclusions from all sections are finally drawn (Section 8).

## 2. UTIs Pathogenesis

A bacterial ethology associated with the rise in faecal microorganisms to the urinary tract is related to most UTIs, but there are other different risk causes, including transmission by person-to-person direct contact and the faecal-oral route [1]. The higher incidence in women is mainly due to their anatomy: the female urethra and the tract to ascend to the bladder, shorter than the male. Besides, anatomical and physiological changes that occur in the urinary tract during pregnancy increase women’s susceptibility to UTIs. The homeostasis of the vaginal microbial ecosystem, formed mainly by lactobacilli, is important to protect vaginal mucosa against the colonization of pathogens and their potential ascending into the urinary system. Therefore, the use of spermicides and antibiotics as well as the menopause period, altering a woman’s vaginal microbiota and decreasing the level of estrogen, favour infections. Incomplete cure or recurrent genitourinary infections also lead to alterations in vaginal microbiota from a predominance of lactobacilli to coliform uropathogens [33]. Moreover, the positive correlation between UTI infection history in first-degree female relatives and UTI risk suggests a genetic component for increased susceptibility [34].

A key step in the pathogenesis of infectious diseases is the pathogenic bacteria adherence to host cells. According to Beerepoot et al. [35], UTIs are settled in three main phases: microbial colonization, bacteria adherence to uroepithelium receptors, and invasion of urinary tract cells. The initial step in the UTIs pathogenesis is the colonization of the urinary tract and the subsequent rise of the pathogen to the bladder to cause cystitis. If left untreated, uropathogens may ascend the ureters to the kidney and establish a secondary infection (acute pyelonephritis). The next phase is the pathogens adherence to uroepithelium, which is mediated by adhesins that recognize the cells’ surface receptors, allowing the uropathogens to withstand the urine hydrodynamic flow. In this manner, uropathogens, such as *Escherichia coli, Klebsiella pneumoniae* and *Staphylococcus saprophyticus,* have the ability to join directly the bladder epithelium causing non-complicated UTIs. However, complicated UTIs are initiated when the bacteria, predominately pathogens such as *Proteus mirabilis*, *Pseudomonas aeruginosa* and *Enterococcus* spp., are attached to a catheter, to a kidney stone or they are retained in the urinary tract by a physical obstruction [8]. Once installed in bladder cells, pathogens internalize, multiply and produce infection. It has been assessed that the uropathogen *E. coli* (UPEC) can access and grow within the bladder cells to form intracellular bacterial communities, which are detected in exfoliated urothelial cells and associated to suffer recurrent UTIs [11]. Moreover, internalized UPEC can persist in quiescence for long periods without causing clinical symptoms [36].

On the other hand, it is worth mentioning that the uropathogens can form biofilms to adhere more easily and increase its resistance. This has an effect on the rest of microorganisms causing greater adherence and multiplication, being able to form thick biofilms that significantly decrease the effectiveness of antimicrobial treatments. When the biofilm resources are limited, mature bacteria are detached and can colonize a new surface to repeat the cycle [37].

## 3. UPEC and Other Uropathogenic Bacteria

The uninfected urinary tract has long been assumed to be sterile in healthy individuals and it was thought that the major uropathogenic organisms only included Gram-negative bacteria, mainly *E. coli* [38,39]. Currently, it is widely held that Gram-positive bacteria either alone or along with Gram-negative uropathogens in the urine are likely to be innocuous. Many species, including some known to cause UTIs, survive and multiply within the urinary tract without causing chronic infection.

*E. coli* is a rod Gram-negative bacterium that colonizes the gastrointestinal tract of humans and animals after birth, forming part of the natural bowel microbiota. In addition, it can be an opportunistic pathogen that uses the intracellular environment to survive and protect itself against the action of antibiotics. Several epidemiological, serological, and bacteriological studies revealed that UPEC is the pathogen most frequently associated with UTIs [40], accounting for three-quarters of all UTIs among outpatients [41,42]. These bacteria have evolved a multitude of virulence factors and strategies that facilitate bacterial growth and persistence within the adverse settings of the host urinary tract.

UPEC’s ability to bind the host uroepithelial receptors is facilitated by virulence factors, hair-like organelles called fimbriae or adhesins that form part of its bacterial wall. These structures, pili or fimbriae, allow for the attachment of *E. coli* in the sites where they usually do not live as the uroepitelum, and the invasion of host cells and tissues to trigger the infection later. Although the biogenesis of bacterial adhesins is a tightly controlled process, UPEC can alter binding features in reaction to environmental changes, including temperature, osmolarity, pH, and nutrient availability [36]. In recurrent UTIs, an increased adherence of *E. coli* to urogenital epithelial cells was seen compared to healthy controls [43].

Different types of *E. coli* adhesins have been described, such as the fimbriae type 1 and type S, both sensitive to the α-d-Mannose receptor, and the fimbriae type P and Afa, resistant to the α-d-Mannose receptor [44]. The most common are the fimbriae type 1 and type P, which mainly play a role in the pathogenesis of cystitis and pyelonephritis, respectively. The fimbriae type 1 has the ability to interact with the receptor α-d-mannose, present in most of the cells of the uroepitelium. In 99% of uropathogenic *E. coli* strains, genes to encode type 1 fimbriae are present [45] and they recognize uroplakin, a FimH-binding transmembrane protein of urothelial lining cells. Mannoside-containing host proteins are encoded by the bacterial backbone DNA and are mainly composed of FimA proteins along with FimF, FimG, and FimH. It is known that Tamm-Horsfall protein (THP) or uromodulin, a highly mannosylated glycoprotein, strongly interacts with FimH of UPEC decreasing interaction with uroplakin. Also, most uropathogenic UPEC strains harbour genes encoding adhesive type1 fimbria but its expression is strictly regulated under physiological growth conditions, increasing its expression when UPEC settles on and infects bladder epithelial cells or colonizes catheters [46]. As a result, UPEC in these sessile populations enhances bladder cell adherence and invasion potential, while UPEC low-expression planktonic populations are subsequently released to the urine. Type P fimbriae, encoded by the gene papG, has been associated with acute pyelonephritis and is prevalent among the strains that cause invasive and persistent UTIs [47].

Other factors of virulence to highlight are the production of toxins, including α-hemolysin (HlyA), cytotoxic necrotizing factor1 (CNF1), secreted autotransporter toxin (SAT), cytolysin A, plasmid-encoded toxin (PET), vacuolating autotransporter toxin (VAT), Shigella enterotoxin-1 (ShET-1) and arginine succinyltransferase (AST) [48], and iron-chelating factors (siderophores) that enables UPEC to capture iron, both mechanisms impair host immune system and obtain nutrients and growth factors necessary for their survival. The expression of these virulence factors converts an *E. coli* commensal strain into an uropathogen, and, often, the inhibition of a single adhesin may cost enough to a bacterium to lose its virulence [40]. Moreover, virulence factors located on a bacterial surface, including capsule and lipopolysaccharides, may also contribute to UTIs providing antiphagocytosis and antibactericidal complement activity [40]. In recent years, the understanding of virulence factors and the behavior of this pathogen has increased remarkably, and may facilitate the application of more precise approaches in phenotypic, molecular diagnosis and the development of alternatives strategies to antibiotics based on antivirulence prophylaxis. Recently, Tamadonfar et al. [9] reviewed the development of small-molecule inhibitors and vaccines targeting virulence factors, as bacterial adhesins or structural components of adherence like siderophores in the effort to reduce UTIs burden and multidrug resistance. Moreover, *E. coli* lineages are more likely to cause UTIs exhibiting an antibiotic resistance phenotype and seem to be more persistent in the rectal tract and pandemic. In this way, the ST131 group of extraintestinal *E coli* strains exhibiting multidrug resistance to beta-lactams and fluoroquinolones and have been identified globally [41].

Besides the most causative agent of UTIs, *E. coli,* other Gram-negative bacteria, such as *K. pneumoniae*, and/or Gram-positive bacteria, including *S. saprophyticus, Enterococcus faecalis*, and group B Streptococcus, are also implied in UTI pathologic outcomes, especially in polymicrobial infections, but they are systematically overlooked [39]. Traditionally, a bacteriological culture of urine is used to isolate and identify UTI pathogens, such as the aerobic fast-growing organisms *E. coli* and *E. faecalis*. Conversely, strains from *Corynebacterium, Lactobacillus* and *Ureaplasma* genera were rarely isolated from the urinary tract as these routine culture techniques are not accurate enough to identify anaerobic slow-growing organisms. However, using advanced detection technologies, these bacteria have been identified as members of the urinary microbiota in multiple studies [38].

It should be remarked that UTI etiologic agents vary according to age, sex, and underlying pathology. Thus, in ambulatory patients, *E. coli* predominates, followed by other Gram-negative bacilli, *Klebsiella* spp. and *Proteus* spp., and Gram-positive cocci, such as *S. saprophyticus, Enterococcus* spp. and *Streptococcus agalactiae*. However, in the elderly infections, *Enterococcus* spp. are the most common agent (11.6%) compared to the rest of the population, which only represent 5.3% of cases [49].

## 4. Cranberry in UTIs Prophylaxis

The increase in global antibiotic resistance and recurrence rates has prompted the exploration and assessment of new therapeutic strategies by antiadhesive components. As a natural alternative, the intake of cranberries as fresh or dried fruits, as well as products derived from them (i.e., juices, extracts, etc.) has been extensively recommended. Cranberry consumption has been indicated to be effective in decreasing the occurrence and severity of UTIs in women and to prevent the adherence of pathogenic bacteria in the urinary tract [13,50,51]. Moreover, it could also decrease UTI related symptoms by suppressing inflammatory cascades as an immunologic response to bacteria invasion [15].

Although numerous epidemiological and intervention studies have proved the efficacy of cranberry products in UTI prophylaxis [16,17], others have shown mixed results [19,52,53]. Therefore, the evidence is still insufficient to define a formal health claim. Apart from differences in composition and doses of the cranberry-based products used in the intervention studies [13], differences among studies have been attributed to the different susceptibility of the UPEC strains to cranberry preventive effects [54]. One of the last meta-analyses concerning this topic reported a large interindividual variability in cranberry efficiency against UTIs, also concluding that patients at some risk of these infections were more susceptible to the beneficial effects of cranberry consumption [4]. Recently, Mantzorou and Giaginis [42] critically analyse the current clinical studies that have evaluated the efficacy of supplementing cranberry products against UTIs in different subpopulations; they conclude that it seems to be prophylactic by preventing infections recurrence; however, it exerts low effectiveness in populations at an increased risk of contracting UTIs. At present, cranberry supplementation can safely be suggested as complementary therapy in women with recurrent UTIs but a lack of cost-effectiveness for cranberry supplementation has been highlighted.

For the most recent decades, flavonoids as proanthocyanidins (PACs), mainly A-type (Figure 1), were assumed to be responsible for these preventive effects against UTIs [21,24]. Nevertheless, this finding has been rebutted due to evidence of the limited absorption of proanthocyanidins and their extensive metabolism by the gut microbiota [22,55,56,57]. In accordance with this, PAC levels in urine after cranberry intake have been found in the low-concentration range (<nM) [58,59]. Moreover, PACs are widely catabolized by the colon microbiota to give bioactive phenolic metabolites that can be further absorbed and also secreted in faeces and urine. Among them, there are single flavonoids, different conjugates (i.e., glucuronides, O-methyl ethers and sulfates) of phenolic acids (i.e., phenylpropionic, phenylacetic, benzoic and cinnamic acids) and other microbial metabolites such as phenyl-γ-valerolactones in urine collected after cranberry intake [50,55,56,57,58,59]. Therefore, these phenolic metabolites might be the responsible compounds behind the preventive actions of flavonoids and phenolic acids present in cranberry on UTIs.

In addition, complex carbohydrates and sugars, terpenes, as well as organic acids such as quinic, malic, shikimic, and citric are other preponderant cranberry phytochemicals. Among them, D-mannose has also been reported to inhibit the adherence of UPEC to uroepithelial cells in vitro, although new studies will certainly be needed to confirm it [38,60]. Vitamin C (ascorbic acid) and fructose have also been suggested as active compounds against UTIs as they promote changes in urine’s physical properties (such as acidification) [11]. Another key characteristic of cranberry juice is the low pH of 2.5 [61]. The unique blend of the organic acids, quinic, malic, shikimic, and citric acid found in cranberries might also lead to antibacterial effects as it has been found in an experimental mouse model of urinary tract infection [62].

## 5. Antiadherence Activity Derived from Cranberry Consumption

The mechanisms implied in the preventive effects of cranberry consumption against UTIs are not completely established and several leading hypotheses have been proposed. Thus, cranberry polyphenols, in particular, their microbial-derived metabolites, are claimed to operate in the phase of bacterial adherence to the uroepithelial cells, disabling or inhibiting the adherence of UPEC and, therefore, preventing bacterial colonization and the progression of UTIs [23,63]. This potential mechanism is depicted in Figure 2. In fact, numerous *ex vivo* studies have confirmed the antiadhesive activity of urine samples collected from volunteers who consumed cranberry products in comparison to urine samples collected from the placebo group. As example of this, a recent study demonstrates the strong ability of human urine after intake of a cranberry chew compared to a placebo chew to inhibit the ex vivo adherence of both P type and type 1 uropathogenic *E. coli* in a randomized, double-blind, placebo-controlled, crossover design pilot trial [20]. In the same way, Baron et al. [59] reported a significant reduction in the adherence and biofilm formation of a *Candida albicans* strain by urine collected after the intake of a cranberry extract and a lecithin formulation with improved oral bioavailability. 

Cell culture methodologies have also applied to assess the antiadhesive activity of cranberry phenolic compounds/fractions/extracts against uropathogens, specifically in bladder epithelial cell lines [23,26,28,64]. It should be noted that some of these in vitro studies have dealt with molecules never appearing or at concentrations far from those found in vivo, and, therefore, the physiological relevance of these studies should be considered with precaution.

Table 1 recompiles comparative data of the antiadhesive activity of different cranberry phenolic compounds and their microbial-derived metabolites against a *E. coli* ATCC^®^ 53503™ into bladder uroepithelial cells (i.e., ATCC^®^ HTB4™ cells) [23,26,28]. As seen from these data, A-type procyanidins (A2 and cinnamtannin B-1) exhibit antiadhesive activity (at concentrations ≥ 250 μM), a feature that is not observed for B-type procyanidins (B2). But, in any case, these concentrations (≥250 μM) are extremely high in comparison to those found in urine (nM range) [62,63], making it unlikely that A-type proanthocyanidins were the compounds responsible for the antiadhesive activity of urine samples collected after cranberry consumption. However, some cranberry-derived phenolic compounds detected in urine after consumption of cranberries and/or cranberry products such as hippuric acid, α-hydroxyhippuric acid, 3,4-dihydroxyphenylacetic acid, and dihydrocaffeic acid 3-*O*-sulfate are able to inhibit the adherence of the uropathogenic bacteria tested at concentrations that may be more physiologically relevant. Additive (and even synergistic) effects among all the cranberry-derived phenolic metabolites present in urine are expected in vivo, which could explain, at least partly, the ex vivo antiadhesive activity of urine samples collected from volunteers who consumed cranberry products [23].

Another interesting finding about the ex vivo capacity to inhibit UPEC adherence after cranberry extract is that of [64] that found, from the data of a cranberry trial with 16 subjects, a positive correlation between urine antiadherence capacity and content in Tamm-Horsfall protein (THP), known as a mechanism of non-specific defense. Authors concluded that the antiadherence effect derived from cranberry consumption not only might be related to the direct inhibition of bacterial adhesins by cranberry-derived metabolites but also to an induction of antiadherence THP in the kidney. Afterwards, they pointed out that cranberry extract stimulates the innate immune defense in the kidney by an increased secretion of THP; on the other hand, the direct inhibition of the bacterial adhesion to host cells has also been proven, which is independent of THP [65]. This is indeed an attractive matter to be further explored.

## 6. Other Potential Mechanisms Involved in Cranberry Efficiency against UTIs

Emerging evidence shows that the gut microbiota has a key role in homeostasis, regulating health and disease at distal sites throughout the body [66]. Although microbial profiles and microbial metabolites of the gut and other organs might influence the urinary microbiota, the relationship between these actively metabolizing organisms and urogenital health is yet to be completely elucidated [38,67]. It is now widely accepted that the urinary tract harbours a complex microbial network which is substantially different from the gut populations [68]. Any imbalance in specific bacterial communities is likely to have a profound effect on urologic health owing to their metabolic output and other contributions. Contrary to the urine of an asymptomatic healthy individual, an altered microbiome with specific dominating urotypes was lately reported in subjects with functional disorders of the urinary tract [67]. On the other hand, non-modifiable host factors (e.g., gender and genetic influences associated with UTI) seem to have a role in the UTIs colonization, probably through its influence in intestinal microbiota [69]. Other host factors, such as the response to fibrinogen depositing at the infection site has been found to be critical to establishing catheter-associated UTIs (CAUTI) [9].

Based on this wide influence of gut microbiota in the body, other hypothesis about the mechanisms of action of cranberry flavonoids and phenolic acids against UTIs is that cranberry components (i.e., A-type proanthocyanidins and their metabolites) would interact with gut microbiota, modulating its composition and/or functionality in such a way that it prevents microbial dysbiosis. In accordance with this hypothesis, a pilot study of the human faecal microbiome (*n* = 10) after 2 weeks of consuming dried cranberries (42 g/day) [70] indicates a shift in the Firmicutes:Bacteroidetes ratio, increases in commensal bacteria, and a particular increase in *Akkermansia* in most subjects. Similarity, in a human trial (*n* = 11) evaluating the effect of a cranberry powder (30 g/day) in the context of an animal-based diet, [71] found that cranberries attenuated the impact of the animal-based diet on microbiota composition, bile acids, and SCFA, evidencing their capacity to modulate the gut microbiota.

More specifically in relation to UTIs, and as it is becoming evident that the intestine is a reservoir for uropathogenic bacteria, in vitro studies have indicated that cranberry flavonoids and phenolic acids might interact with extra-intestinal *E. coli* (ExPEC) and decrease its (transient) intestinal colonization, consequently reducing the risk of UTI incidence [72]. This potential mechanism of action of cranberry polyphenols against UTIs is also depicted in Figure 2.

Therefore, the effects of cranberry polyphenols as flavonoids and phenolic acids against UTIs would be influenced by what has been called a “two-way interaction” between polyphenols and gut microbiota [73] in the sense that not only is the gut microbiome involved in the metabolism of cranberry polyphenols, but also the cranberry compounds and their microbial metabolites may impact on gut microbiota and induce preventive effects on intestinal colonization by uropathogens. This two-way interaction between polyphenols and intestinal bacteria has been reviewed focusing on other foods such as tea [74] and wine polyphenols [75].

In other words, the beneficial effects of cranberry against UTIs would be modulated by our intestinal microbiota through its different capacity in metabolizing cranberry flavonoids and phenolic acids into bioactive metabolites. This is the case, for example, in phenyl-γ-valerolactones, one of the most abundant metabolites in the bioactive urine fraction after cranberry intake [56,59] and whose antiadhesive activity has been proven in vitro [28]. Therefore, the inter-individual differences found in the preventive effects of cranberry against UTIs may be attributed, at least partly, to the differences in intestinal microbiota composition among individuals.

## 7. Combined Action of Cranberry and Probiotics against UTIs

The vaginal microbiota, that has predominance of lactobacilli, plays a dynamic and often critical role in UTI pathology by maintaining a low pH and avoiding uropathogen colonization by competitive exclusion. However, hormonal change due to estrogen deficiency, antimicrobial therapy, contraceptives, incomplete healing and recurrence of UTIs lead to a shift in the local microbiota (dysbiosis), increasing the risk of pathogens triggering pathologies (UTIs, bacterial vaginosis and yeast vaginitis) [69,76]. Therefore, the instillation of lactobacilli in the vaginal cavity is presented as a potential therapy for urinary tract care [29,77]. After their oral administration, selected strains of *Lactobacillus* have been found in vaginal exudates and faecal samples [30,78]; *Lactobacillus* translocation from the intestinal mucosa to distal mucosal surfaces has also been reported [79].

The use of probiotics/lactobacilli against UTIs has been assessed [80]. The inhibition of the adherence of pathogenic bacteria (i.e., UPEC and others) to epithelial vaginal cells by different *Lactobacillus* strains has been proven in many in vitro studies [81,82,83]. As a recent example, González de Llano et al. [26] reported the inhibitory effects of different strains of *Lactobacillus salivarius*, *Lactobacillus plantarum* and *Lactobacillus acidophilus* on the adherence of various uropathogenics strains from *E. coli*, *Staphylococcus epidermidis* and *E. faecalis* to bladder cells T24. However, the mechanisms behind these inhibitory effects have thus far not been elucidated, with most studies relying on inferred evidence [30,84,85]. The production of biosurfactants, bacteriocins, lactic acid and hydrogen peroxide by *Lactobacillus* spp. seems to inhibit UPEC growth and adversely affects its fimbrial structure and adherence as well as upregulate immunogenic membrane proteins [86,87]. Moreover, the phenolic biotransformation process by the microbiota or probiotic bacteria into low molecular weight cranberry phenolic metabolites can enhance their pharmacological activities by generating additional biologically active metabolites. Recently, the role of *L. rhamnosus* probiotic in the degradation of flavonoids which are present in cranberry pomace fractions, and the effect of the yielded cranberry metabolites on hepatocellular carcinoma HepG2 cells in vitro have been investigated by Rupasinghe et al [88] and they found an inhibitory effect on HepG2 cell proliferation in a dose- and time-dependent manner enhancing their anticancer activity.

Several human intervention studies have evaluated the effectiveness of specific *Lactobacillus* strains’ consumption to prevent or treat UTIs [31,35,89]. So far, the results of these clinical studies remain inconclusive as a consequence of the wide diversity of populations, small sample sizes, different probiotic species and dosage forms such as vaginal and oral, and the use of incorrect dosing strategies. Therefore, more randomized clinical trials about the effect of probiotics against UTIs should be conducted to make a stronger recommendation as a probiotic benefit cannot be ruled out [11,35].

New products claiming protection against UTIs and combining cranberry extracts and *Lactobacillus* probiotic preparations have recently been lunched onto the market, being available to the general public without prescription. Many of these products do not have scientific support, although some human studies on the combination of cranberry–probiotics have been reported. In this way, Montorsi et al. [32] have proved, in a pilot study, the efficacy of the combined intake of *Lactobacillus rhamnosus* SGL06, cranberry and vitamin C for the prevention of recurrent UTIs. Polewski et al. [90] have also evaluated the impact of lactobacilli in combination with cranberry A-type proanthocyanidins on reducing the invasiveness of extra-intestinal pathogenic *E. coli* (ExPEC) to intestinal cells. The exploration and incorporation of advanced probiotic formulations and microbiome-targeted treatment strategies within urology is warranted [38].

Lastly, new therapies based on vaccination and small-molecule inhibitors targeting uropathogens virulence factors, are being investigated as future treatments against UTIs [9]. In this lane, Ahumada-Cota et al. [91] have recently outlined the potential treatment and control of recurrent UTIs in adults with autologous bacterial lysates, and its composition shown that different surface components of *E. coli*, mainly outer membrane proteins are potential immunogens. Thus, bacterial lysates could be potential candidates to create a polyvalent protective vaccine against recurrent *E. coli*-associated UTIs. Alternatively, the development of a synthetic urinary microbiota for transplantation might lead to an effective treatment for patients with recurrent UTIs, but still all these new approaches are under development.

It is worthy of note that cranberries, like some traditional medicines, are a well-known functional food, whose efficacy has been assayed in clinical trials. However, the compounds directly responsible for many of cranberries’ reported health benefits remain unidentified. They are complex mixtures whose constituents have a synergistic effect by acting at different levels on multiple targets and pathways or reducing toxicity and drug side effects [92]. Nowadays, cranberry characterization and research into single/pure compounds focusing on synergy among pure compounds are essential for understanding their therapeutic effect [92]. In this line, Coleman and Ferreira [93] have recently reported that complex carbohydrates, specifically xyloglucan and pectic oligosaccharides, are significant components of cranberry products, and a new target for focused investigations of possible prebiotic effects that contribute to cranberry bioactivity. However, the use of refined cranberry oligosaccharide constituents *in vivo* may be impractical and ineffective until more is understood about the various mechanisms by which different cranberry components work together to influence overall health.

## 8. Conclusions

The consumption of cranberry (*Vaccinium macrocarpon*) has been extensively recommended for UTI prophylaxis and the relief of adverse symptoms. *In vivo* evidence has been translated to *in vitro* verification, resulting in the widely recommended use of cranberry juice in traditional medicine for urinary tract pathologies. This review summarizes some relevant considerations about cranberry polyphenols as flavonoids and phenolic acids, and UTIs derived from the last scientific publications. One of our first conclusions is that, besides uropathogenic *Escherichia coli* (UPEC), other etiological agents such as *Klebsiella pneumoniae* or the Gram-positive bacteria of *Enterococcus* and *Staphylococcus* genera seem to be widely involved in UTIs. In particular, complicated UTIs caused by UPEC and *Enterococcus* spp. represent a major health care concern nowadays. The application of advanced techniques for UTI diagnosis comprising PCR, microarray and next-generation sequencing, together with the study of the individual urological microbiome, may improve diagnosis and treatment of these infections. The other outcome of this review is that the existing clinical trials support substantial evidence about the use of cranberry products as total or partial therapeutic alternatives to antibiotics in UTIs, although it has also been seen that cranberry effectiveness is individual- and/or case-dependent. This variability among individuals and cases has been attributed to different manufactured cranberry-based products and doses, as well as to the lack of systematic protocols for the selection of subjects and clinical assays. However, as more is known about the metabolism of cranberry polyphenols, new factors such as host microbiota, endogenous metabolism, host genetics and the immune system may also play a role in cranberry effectiveness against uropathogens. A question that remains to be fully answered is which cranberry and/or cranberry-derived components are mainly responsible for its protective effects against UTIs. Although recent studies have shed light on the antiadherence activity derived from cranberry consumption, other mechanisms might be jointly implicated that need further investigation. Knowledge of these mechanisms (and the cranberry components involved) might result in a better management of the potential of cranberry products against infections. For instance, nutraceuticals and other forms containing cranberry bioactive compounds and ensuring optimum bioavailability and effectiveness could be designed. Finally, pieces of evidence regarding treatment against UTIs with probiotics and their combinations with cranberry are too limited to draw any conclusions but they seem to indeed be a promising strategy for future treatments. However, any development must be supported by exhaustive research specific to the strain/s and cranberry products, as well as about the mechanisms involved in these potential beneficial effects of probiotics and their combinations with cranberry.

## Figures and Tables

**Figure 1 molecules-25-03523-f001:**
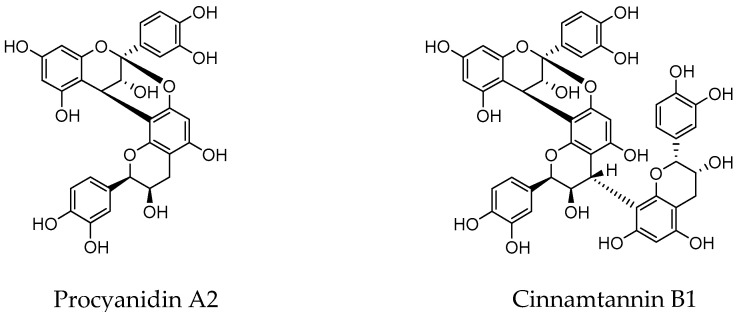
Structures of A-type proanthocyanidins found in cranberry.

**Figure 2 molecules-25-03523-f002:**
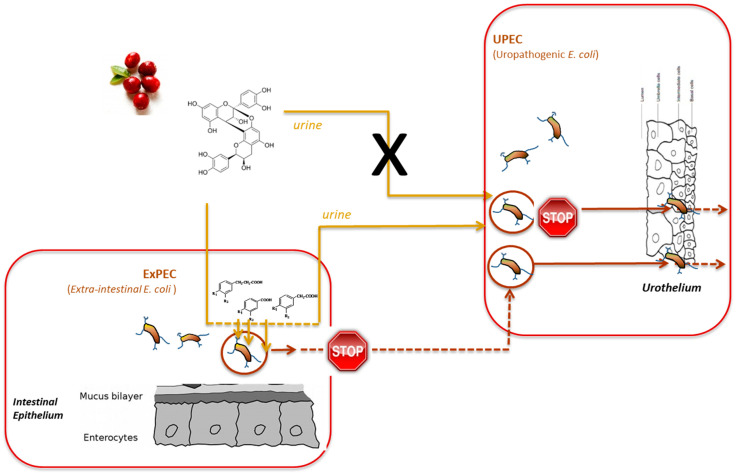
Proposed mechanisms of cranberry polyphenols action against urinary tract infections (UTIs).

**Table 1 molecules-25-03523-t001:** Adherence Inhibition (%) of the of *E. coli* ATCC^®^ 53503™ to ATCC^®^ HTB4™ cells by phenolic compounds (Adapted from González de Llano et al. [23,26,28]).

	Concentration (µM)		
	100	250	500
***Flavan-3-ols***			
Cinnamtannin B1	1.05	4.11	13.95 *
Procyanidin A2	23.67	30.7	54.5 **
Procyanidin B2	6.79	10.0	−14.7
(™)-Epicatechin	−6.02	−1.21	−5.82
**Simple phenols**			
1,2-Dihydroxybenzene (catechol/pyrocatechol)	17.0 *	26.0 *	33.2 **
1,3,5-Trihydroxybenzene (phloroglucinol)	−8.53	17.6	−8.15
**Benzoic acids**			
Benzoic acid	16.5 *	23.3 **	32.2 **
3-Hydroxybenzoic acid	11.1	17.0*	−9.7
3,4-Dihydroxybenzoic acid (protocatechuic acid)	25.5 *	24.0	9.44
4-Hydroxy-3-methoxybenzoic acid (vanillic acid)	18.3 **	24.9 **	29.2 **
3,4,5-Trihydroxybenzoic acid (gallic acid)	−3.72	19.7	40.6**
**Phenylacetic acids**			
Phenylacetic acid	33.5 *	39.0 **	40.6 **
3-Hydroxyphenylacetic acid	15.0	11.9	19.4
3,4-Dihydroxyphenylacetic acid	18.6	32.5 *	37.0 **
4-Hydroxy-3-methoxyphenylacetic acid	7.11	11.92	12.8
**Phenylpropionic acids**			
3-Phenylpropionic acid	−11.8	14.7	12.2
3-(3-Hydroxyphenyl)-propionic acid	10.2	18.6	30.5 *
3-(3,4-Dihydroxyphenyl)-propionic acid	6.66	1.19	13.1
3-(3,4-Dihydroxyphenyl)-propionic acid 3-*O*-sulphate sodium salt	6.52	11.22	21.0 *
**Dihydroxyphenyl-γ-valerolactones**			
5-(3′,4′-Dihydroxyphenyl)-γ-valerolactone	6.79 ± 3.92	9.95 ± 8.28	19.4 ± 10.3 *
5-Phenyl-γ-valerolactone-3′,4′-di-*O*-sulphate	−0.22 ± 0.71	14.7 ± 1.5	30.3 ± 3.6 **
5-(4′-Hydroxyphenyl)-γ-valerolactone-3′-*O*-sulphate	11.9 ± 1.7	10.2 ± 3.9	22.2 ± 5.9 **
5-(3′-Hydroxyphenyl)-γ-valerolactone-4′-*O*-sulphate	10.1 ± 3.1	16.1 ± 6.1*	24.2 ± 3.1 **
**Hippuric acids**			
Hippuric acids	15.6	14.9	25.5 *
α-Hippuric acid	20.8	23.01	20.0

* Mean significantly different from zero (*p* < 0.05) using a one-sample t-test. ** Mean significantly different from zero (*p* < 0.01) using a one-sample t-test.

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
