# Peer review of "Cranberry Polyphenols and Prevention against Urinary Tract Infections: Relevant Considerations"

_molecules, 2020, doi:10.3390/molecules25153523_

Round 1

Reviewer 1 Report

The review reports and discusses the potential preventive activity of Cranberry polyphenols against urinary tract infections.

The paper is well organized and reports a good amount of data to support the topic addressed.

However, it is important to add in the manuscript the amount of cranberry polyphenols used in the cited data (administered/consumed/used for in vitro). In some references are used high quantities of these compounds and this should be commented critically in light of the problems of bioavailability, the role of their metabolites and / or toxicity of these molecules. In this contest, the author write “PACs levels in urine after 237 cranberry intake have been found in the low-concentration range (<nM)” (line 236), but successively they report at line 278 and in table 1 “A-type procyanidins (A2 and cinnamtannin B-1) exhibit antiadhesive activity (at concentrations ≥250 μM)”. This latter data obtained by using concentrations of molecules that are extremely higher than those bioavailable should be critically considered and commented.

The use of an extract have strengths (combined/synergistic effects of the wide range of molecules present in the extracts, pleiotropy…) and weaknesses (Not-specific effect … ) that must be described/commented by the authors

I suggest to add a short paragraph (or add in discussion section) about the possibility to isolate from Cranberry the most bioactive compounds and utilize them as nutraceuticals

Author Response

Response Reviewer 1. Comments

The review reports and discusses the potential preventive activity of Cranberry polyphenols against urinary tract infections. The paper is well organized and reports a good amount of data to support the topic addressed.

Point 1: However, it is important to add in the manuscript the amount of cranberry polyphenols used in the cited data (administered/consumed/used for in vitro). In some references are used high quantities of these compounds and this should be commented critically in light of the problems of bioavailability, the role of their metabolites and / or toxicity of these molecules. In this contest, the author write “PACs levels in urine after 237 cranberry intakes have been found in the low-concentration range (<nM)” (line 236), but successively they report at line 278 and in table 1 “A-type procyanidins (A2 and cinnamtannin B-1) exhibit antiadhesive activity (at concentrations ≥250 μM)”. This latter data obtained by using concentrations of molecules that are extremely higher than those bioavailable should be critically considered and commented

Response 1: We thank the reviewer for his/her comments with which we agree.

Effectively, in the bibliography there are numerous studies (in vitro studies, in particular) in which molecules never appearing or at concentrations far from those found in vivo, have being used. We think that what the reviewer suggests about including the amount of cranberry polyphenols used in the cited references would be quite confusing and useless, but we have now included some new statements reflecting this reviewer´s comment (lines 270-272):

“It should be noted that some of these in vitro studies have dealt with molecules never appearing or at concentrations far from those found in vivo, and, therefore, the physiological relevance of these studies should be considered with precaution.”

A new statement has also included to reflect the reviewer´s comment about the high concentration of A-type procyanidins (≥250 μM) needed to exhibit antiadhesive activity in vitro in comparison their concentration in urine (nM range) (lines 277-280):

“But, in any case, these concentrations (≥250 μM) are extremely higher in comparison to those found in urine (nM range) [62][63], making it unlikely that A-type proanthocyanidins were the compounds responsible for the antiadhesive activity of urine samples collected after cranberry consumption. However, some…”

Point 2: The use of an extract have strengths (combined/synergistic effects of the wide range of molecules present in the extracts, pleiotropy…) and weaknesses (Not-specific effect … ) that must be described/commented by the authors.

Response 2: Please, note that this fact had already been commented in the manuscript (lines 290-292), but, according to the reviewer’s suggestion, we have added the above paragraph at the end of discussion (lines 395-406).

“It is worthy of note that cranberry like some traditional medicines are well-known functional food, whose efficacy has been assayed in clinical trials but compounds directly responsible for many of its reported health benefits remain unidentified. They are complex mixtures whose constituents have a synergistic effect by acting at different levels on multiple targets and pathways, or reducing toxicity and drug side effects [96] [97]. Nowadays, cranberry characterization and research into single/pure compounds focusing on synergy among pure compounds are essential for understanding therapeutic effect of them [96]. In this line, Coleman & Ferreira [97] have recently reported that complex carbohydrates, specifically xyloglucan and pectic oligosaccharides, are significant components of cranberry products and a new target for focused investigations of possible prebiotic effects that contribute to cranberry bioactivity. However, the use of refined cranberry oligosaccharide constituents in vivo may be impractical and ineffective until more is understood about the various mechanisms by which different cranberry components work together to influence overall health”.

  1. Yuan, H.; Ma, Q.; Cui, H.; Liu, G.; Zhao, X.; Li, W.; Piao, G. How Can Synergism of Traditional Medicines Benefit from Network Pharmacology? Molecules 2017, 22, 1135.
  2. Coleman, C.M.; Ferreira, D. Oligosaccharides and Complex Carbohydrates: A New Paradigm for Cranberry Bioactivity. Molecules 2020, 25, 881.

Point 3: I suggest to add a short paragraph (or add in discussion section) about the possibility to isolate from Cranberry the most bioactive compounds and utilize them as nutraceuticals

Response 3: According to the reviewer´s comment about isolation of cranberry active compounds, a new sentence has now been included in the Conclusions section (lines 431-433):

“For instance, nutraceuticals and other forms containing cranberry bioactive compounds and ensuring optimum bioavailability and effectiveness, could be designed.”

Reviewer 2 Report

The review article describes current state of knowledge for the role of cranberry juice and its components in prevention of urinary tract infections. The review is very well organized, well written and lists all the evidence, sometimes anecdotal, of beneficial role cranberry juice in infection prevention. The review does not explain molecular mechanisms of the effects except the proposed view of others. Since it is a general review, the approach is acceptable.

Typographical errors:

Line 161: should be "interact" instead of "act"

Author Response

Response Reviewer 2. Comments

The review article describes current state of knowledge for the role of cranberry juice and its components in prevention of urinary tract infections. The review is very well organized, well written and lists all the evidence, sometimes anecdotal, of beneficial role cranberry juice in infection prevention

Point 1: The review does not explain molecular mechanisms of the effects except the proposed view of others. Since it is a general review, the approach is acceptable.

Response 1: We thank the reviewer for his/her comments. Certainly, molecular mechanisms of the effects were reported according to authors ‘findings in their published researches which we summarize in this review as the mechanisms that are behind the UTI-protective effects of cranberry are still unraveled. However, we also hypothesize about them.

Point 2: Typographical errors: Line 161: should be "interact" instead of "act"

Response 1: Thank you for noticing, it has been corrected.